# Mechanical Stimulation of Red Blood Cells Aging: Focusing on the Microfluidics Application

**DOI:** 10.3390/mi16030259

**Published:** 2025-02-25

**Authors:** Alexander Gural, Ivana Pajić-Lijaković, Gregory Barshtein

**Affiliations:** 1Blood Bank, Hadassah-Hebrew University Medical Center, Jerusalem 91120, Israel; gural@hadassah.org.il; 2Department of Chemical Engineering, Faculty of Technology and Metallurgy, University of Belgrade, 11000 Belgrade, Serbia; iva@tmf.bg.ac.rs; 3Department of Biochemistry, The Faculty of Medicine, Hebrew University, Jerusalem 91120, Israel

**Keywords:** red blood cells, mechanical stress, RBC features, artificial organs, RBC fatigue

## Abstract

Human red blood cells (RBCs) are highly differentiated cells, essential in almost all physiological processes. During their circulation in the bloodstream, RBCs are exposed to varying levels of shear stress ranging from 0.1–10 Pa under physiological conditions to 50 Pa in arterial stenotic lesions. Moreover, the flow of blood through splenic red pulp and through artificial organs is associated with brief exposure to even higher levels of shear stress, reaching up to hundreds of Pa. As a result of this exposure, some properties of the cytosol, the cytoskeleton, and the cell membrane may be significantly affected. In this review, we aim to systematize the available information on RBC response to shear stress by focusing on reported changes in various red cell properties. We pay special attention to the results obtained using microfluidics, since these devices allow the researcher to accurately simulate blood flow conditions in the capillaries and spleen.

## 1. Introduction

Human red blood cells (RBCs) are highly specialized cells that have lost all organelles and most intracellular machinery during maturation. RBCs play a vital role in nearly all basic physiological processes. They are the key cells responsible for gas exchange. Due to their elasticity, RBCs can deform and pass through tiny capillaries in the blood circulation. While flowing within the bloodstream, RBCs are subjected to varying levels of shear stress. The typical shear stress levels in different vessels are as follows: aorta 0.1–2.2 Pa [1], arteries 1.0–7.0 Pa [2], veins 0.1–0.6 Pa [3], and capillaries 0.3–9.5 Pa [4]. Additionally, these values can rise to 10 Pa during increased cardiac output or hypertension, reaching as high as 15 Pa in conjunctival pre-capillary arterioles or 20 to 50 Pa in an artery with a stenotic lesion [5].

During blood circulation in vivo, the cells experience the highest mechanical stress (MS) when passing through the submicronic splenic inter-endothelial slits (IES) [6,7]. When squeezing through the thin IES (width of 0.25–1.2 μm; length of 0.9–3.2 μm and depth of 5 μm) [7,8,9] between adjacent venous endothelial cells, RBCs must undergo extensional deformation [10,11]. The RBC passes the slits of the red pulp of the spleen approximately every 200 min [9,12], and it takes about 0.01–0.1 s for a cell to pass through a slit [6,9,10] under pressures of hundreds of Pa [6,13].

In modern medicine, the use of artificial organs has become widespread. An artificial organ may be defined as a human-made device designed to replace, duplicate, or augment, functionally or cosmetically, a missing, diseased, or otherwise incompetent part of the body [14]. Some of these devices are designed to ensure adequate blood circulation (artificial heart, valves, blood vessels, etc.), to allow for gas–blood exchange (heart–lung machine), and to remove extra fluid and waste products from the blood (artificial kidney). In many situations, blood flow in artificial organs is associated with the brief exposure of RBCs to very high levels of shear stress, even up to hundreds of Pa [5,15,16].

As a result of being subject to significant mechanical stress (MS), some properties of the cytosol, the cytoskeleton, and the cell membrane may be significantly affected [17]. Under certain conditions, RBCs can be destroyed (hemolyzed) [18,19,20,21,22,23,24]. Functional and structural changes in RBCs are enhanced in response to supra-physiological (i.e., exceeding the typical values maintained in the human body) exposure to shear stress [15,19,25,26,27]. Previous studies in which the authors examined the changes occurring in RBCs due to MS considered only a narrow range of cell features. Primary attention has been paid to cell hemolysis [15,28,29,30], a change in the RBC morphology [31,32,33,34], deformation [5,15,20,21,35], and cell fragility [36]. Separate studies have also established that exposure of cells to MS may lead to ATP depletion [32], phosphatidylserine (PS) externalization [37], vesiculation [34,37,38,39] membrane blebbing [17,26], cell fragmentation [40,41,42,43], alteration of protein composition [39], elevated concentration of intracellular calcium [44,45], increased membrane oxidation [46], a decrease in the cell surface charge [36], elevation of RBC aggregation [36], and RBC adhesion to endothelium [47]. Ultimately, these changes lead to the acceleration of RBC aging and destruction [19,48,49,50,51,52,53]. The impact of RBC exposure to MS was analyzed in detail using theoretical models [52,54,55] and in vitro and ex vivo experiments [19,49,50,53,56,57].

Freitas Leal et al. [37] demonstrated that the circulation of stored RBCs through a heart–lung machine induces physiologically significant changes in the RBC structure. These changes include increased osmotic fragility, deformability and aggregation, and alterations in the externalization of PS and microvesicle generation. The degree and kinetics of these changes depend on the RBC’s storage time.

Watanabe et al. [42] used a shear (counter-rotation) system coupled to a microscope to directly monitor RBC fragmentation and hemolysis, identified in real time after 40 s exposure to 288 Pa. Later, the same research group [17] visualized the presence of abnormal RBCs (with damaged morphology) under prolonged (more than 100 s) exposure to shear stress of 60 Pa that induced asymmetric cell elongation. Furthermore, using numerical simulations, Zhu et al. [6] demonstrated that exposure of RBCs to MS in the splenic inter-endothelial slits can contribute to cell membrane vesiculation and a decrease in its surface area-to-volume ratio. The observed changes, in turn, lead to an alteration in the cell’s functionality, for example, a reduction in its deformability [58,59]. Thus, the generation of mechanical stress exerted on cells that cyclically move through microcapillaries and narrow lumens represents a fundamental element in cell maturation and aging [34].

The above data convincingly demonstrate that mechanical loading of RBCs leads to changes in a wide range of features and can be associated with RBC aging [7,60]. Below, we will describe how some cell properties change under MS exposure and attempt to systematize the available information on RBC response to shear stress by focusing on RBC structural changes. It is not our aim to address the impact of the strength and duration of the exposure to stress on its outcome, as this issue has been considered in detail in several experimental and numerical studies. Similarly, we will not discuss the hemolysis of the RBCs subjected to MS, as discussed in several reviews. Instead, we will focus on the effects of mechanical stress on the features of RBCs that retain their integrity in the bloodstream following exposure.

We would also like to note that as different research groups study the impact of numerous devices (either prevalent in clinical settings or specifically designed to simulate mechanical stress in vitro) on the properties of the red cells, their detailed description would require a separate review. For our purpose, it is sufficient to mention only those that will be mentioned in this review (see Table 1 below). Table 1: Brief information about the devices for MS generation. We pay special attention to the results obtained using microfluidics.

## 2. Red Blood Cells’ Functionality and Structure

Red blood cells (RBCs) play a crucial role in the circulatory system, being primarily responsible for transporting oxygen and essential nutrients to various tissues while also facilitating the removal of metabolic byproducts. Research indicates that RBCs have a relatively consistent lifespan within the bloodstream, typically averaging 110 days [73]. Nevertheless, the lifespan of individual RBCs can exhibit considerable variability. In healthy individuals, the average lifespan of a RBC is approximately 115 days, although it may fluctuate between 70 and 140 days [74,75]. Thus, Cohen et al. reported that the average age of circulating RBCs in healthy subjects ranged from 38 to 60 days [74].

### 2.1. Basic Characteristics of RBCs and Their Functionality

RBCs are distinct from most human cells because they lack complex internal structures like nuclei or mitochondria. Instead, their cytosol contains between 27 and 33 picograms of hemoglobin per cell, a protein responsible for transporting oxygen. Healthy RBCs exhibit a biconcave morphology (see Figure 1), characterized by a round disk shape with a diameter of approximately 7 to 8 μm and a thickness of around 2.0 μm, resulting in a median cellular volume ranging from 80 to 100 femtoliters.

The shape of RBCs undergoes significant changes during the aging process and can also alter in response to various pathological conditions or while stored in a blood bank.

The RBC comprises two primary elements: the membrane and the cytosol. Hemoglobin (Hb), the main protein found in the cytosol, plays a vital role in the reversible binding and transport of oxygen, hydrogen, and carbon dioxide. Hb is structured as a tetramer, consisting of four subunits known as globin chains, each associated with an iron atom part of a heme molecule. Consequently, each hemoglobin molecule can transport up to four oxygen or carbon dioxide molecules.

Given a specific volume and surface area, the RBC membrane must maintain a minimum elastic energy to preserve the cell’s discoid (biconcave) shape. This membrane comprises a complex assembly of lipids, proteins, and carbohydrates that interact to form a dynamic and fluidic structure, organized into a lipid–protein bilayer supported by a spectrin-based cytoskeleton (Figure 2). In terms of dry weight, the composition of the RBC membrane is characterized by a protein-to-lipid-to-carbohydrate ratio of 49:43:8 [76]. Unlike simple lipid vesicles, the RBC membrane exhibits semi-solid characteristics, with both elastic and viscous properties. The fluid mosaic model, proposed by Singer and Nicholson [77], conceptualizes the erythrocyte membrane as a two-dimensional fluid-like lipid bilayer interspersed with proteins [78]. The elasticity of the bending in the bilayer contributes to its viscoelastic characteristics [79].

### 2.2. General View of the Structure of the Erythrocyte Membrane

Below, we will examine several membrane factors in greater detail: lipids, band 3 (one of the main membrane proteins), the cytoskeletal network, and hemoglobin, which is attached to the inner surface of the membrane. Our selection is based on the fact that these components play a crucial role in the cell’s ability to endure external mechanical stress.

Lipids: In red blood cells (RBCs), lipids are essential components of the membrane, with their hydrophobic tails facing inward and hydrophilic polar head groups oriented toward either the exterior (extracellular) or the interior (cytosolic) surface (Figure 2). Three types of lipids are present in the RBC membrane: phospholipids (50%), cholesterol (40%), and glycolipids (10%). Each type serves a specific function. The RBC membrane is characterized by an asymmetrical distribution of phospholipids between the two sides of the bilayer. The inner leaflet mainly contains aminophospholipids (phosphatidylethanolamine and phosphatidylserine), while the outer leaflet predominantly contains neutral phospholipids (phosphatidylcholine and sphingomyelin). The presence of PS, which carries a negative charge, on the inner monolayer creates a significant charge difference between the two sides of the bilayer. When a cell is exposed to oxidative or mechanical stress, the asymmetric distribution of phospholipids in the membrane is disrupted, leading to the externalization of PS. The externalization of PS on the RBC surface leads to changes in various cell properties, such as morphology and interactions with other cells. Moreover, the externalization of PS on the membrane surface also acts as a marker for “eat me”.

Band 3 molecules are transmembrane proteins whose hydrodynamic radius is approximately 7.6 nm, while the tetramer has a radius of 11 nm at room temperature and pH 7.2 [80]. The total number of band 3 molecules in a single erythrocyte is about 1 × 106. Thermal fluctuations of the erythrocyte membrane can induce conformational changes in the cytoplasmic domain of band 3, resulting in electrostatic interactions between highly anionic N-terminal domains of these molecules, thereby intensifying their short-range self-associative tendency [80]. Positive hydrophobic mismatch effects promote a long-range self-associative tendency [81,82]. Band 3 clustering is especially pronounced under hypotonic conditions [83].

Cytoskeleton: The actin cortex behaves as a viscoelastic solid [83]. In erythrocytes, the mesh size of the actin cortex ranges from 80 to 100 nm [84], while its thickness is approximately 2 nm [85]. Notably, the surface shear modulus of the erythrocyte cortex is significantly lower than that of the fibroblast cortex, indicating a more elastic structure. Specifically, the shear modulus of the erythrocyte cortex is reported to be 5.7 × 10^−6^ Pa [85], in contrast to the fibroblast shear modulus, which ranges from 2 to 2 × 10^−3^ Pa [86]. The viscoelastic properties of the cortex are influenced by two primary factors: (i) the flexibility of spectrin filaments and (ii) the coupling between the cortex and the bilayer. The flexibility of spectrin filaments is determined by the ratio of the filament contour length to its persistent length. The contour length of spectrin is approximately 200 nm [84], while its persistent length ranges from 15 to 25 nm [87], indicating that these filaments are relatively flexible. However, the mobile fraction of band 3 molecules forms low-affinity complexes with spectrin filaments, which increase the persistent length and consequently reduce the flexibility of spectrin [83]. In contrast, actin filaments exhibit behavior similar to rigid rods. Additionally, the coupling between the bilayer and the cortex affects the bending of the bilayer [83].

Membrane bounding hemoglobin: The membrane-associated fraction of hemoglobin (Hb) ranges from 0.5% to 12% of the total cellular Hb content [10]. The hemoglobin molecule is nearly spherical, with an average hydrodynamic radius of about 3.2 nm [88]. Research carried out by Ross and Minton [89] indicated that Hb solutions with concentrations up to 5.2 mol/m^3^ can be considered suspensions of hard spherical particles. The diffusion coefficient of Hb at infinite dilution is measured at 6.4 × 10^−11^ m^2^/s [88]. The isoelectric point of HbA (consisting of 2α and 2 β globin chains) is 6.8 [88], while HbA2 (consisting of 2α and 2σ globin chains) has an isoelectric point of 7.4 [90]. This indicates that at a pH of 7, HbA2 molecules act as cations and migrate toward the negatively charged erythrocyte membrane, consistent with the zeta potential of the erythrocyte membrane, which is approximately −15 eV [91]. Various mechanisms of Hb interaction with erythrocyte membranes have been documented, including (a) electrostatic binding of deoxy-Hb to the cytoplasmic domain of band 3 protein (**anion exchanger-1,** AE-1), (b) covalent crosslinking to membrane components via disulfide bonds, and (c) adsorption onto membrane lipids through hydrophobic interactions [92].

## 3. Alteration of RBCs’ State Following Their Exposure to Mechanical Stress

### 3.1. ATP Depletion

In cell biology, adenosine triphosphate (ATP) is known as an energy carrier. However, in the last decade, researchers have increasingly focused on its crucial role in physiological signaling, including blood microcirculation [93]. This role is closely connected with ATP release from an RBC to the blood under shear stress due to cell deformation. Sprague et al. [94] formulated this cause-and-effect relationship as follows; the increased shear stress inside narrower channels causes the RBCs to deform, and the deformation triggers the release of ATP. After 35 years of research, we can say that the process of ATP release depends on several factors, such as the (i) diameter of blood vessels, (ii) blood flow rate, and (iii) viscoelasticity of RBC membrane. Different authors suggest controversial mechanisms of mechano-sensitive ATP release from RBC, and several hypotheses have been put forward to explain this phenomenon [84,95,96]; what is certain is that the release of ATP is initiated at low shear stress values and occurs within a short time after the onset of exposure [59,93].

Thus, Wang and his colleagues [59] studied in detail the kinetics of ATP release during the flow of RBC suspension through microchannels. Their results show that two different time scales are associated with stress-induced ATP release from RBC; during the first activation stage, characteristically lasting 3–6 ms, the deformation of cells does not lead to ATP release, which only occurs during the second stage (during 25 - 75 ms) following an exposure. In this case, the time interval depends on the magnitude of the shear stress but is insensitive to the rigidity of the RBC membrane. A subsequent decline in cellular ATP levels [35] results in RBC fatigue, morphological alterations, and a diminished capacity for deformation, ultimately culminating in hemolysis [97].

### 3.2. RBC Vesiculation

RBC vesiculation is one of their primary cell aging mechanisms [8,98]. This process induces disintegration of the lipid bilayer. To model this process in vitro, various research groups productively used rheometers (for cycling exposure) [13,21,27,52] or different constructions of microfluidic devices [35,57,58,99,100,101]. Specifically, two types of microfluidic devices were used: one creating a constant level of impact [59] and another making a cyclical exposure of the flowing cell [35,57,58]. Comparing the experimental results obtained under different MS conditions, the authors concluded that cyclical impact (oscillatory shear stress) causes more significant vesicle formation [35,57,58]. Numerical methods were also successfully employed as an addition to the experimental analysis of vesicle formation following MS exposure [1,102].

Under physiological conditions, the most significant formation of vesicles occurs during blood filtration through the spleen, when RBCs are squeezed through the IES (0.65 μm long and 2–3 μm high). Several research groups have attempted to study vesicle formation under the influence of splenic blood flow using numerical simulations of RBC flow through human splenic IES. One of the main conclusions made by the authors was that RBC squeezing through IES induces partial disintegration of some protein complexes, leading to a decrease in the lipid bilayer–cytoskeleton attachment, which stimulates the release of membrane components such as vesicles [1,35,102]. Zhu et al. [1] conclude that the formation of vesicles under conditions of MS is stimulated by the binding of degraded hemoglobin to the inner surface of the membrane. It is essential to emphasize that MS-generated vesicles are not formed by cell rupture [103]. In addition, the literature widely discusses the mechanism of MS-stimulating RBC vesiculation [8,35,103] and considers the role of Ca^2+^ influx in this process [103].

Li et al. [102] tried to understand the mechanism of RBC vesiculation following mechanical stress. The authors analyzed the response of two types of RBCs (hereditary spherocytosis and hereditary elliptocytosis) to their passage through IES [102]. These two types of RBCs showed quite different behavior in the context of the cytoskeleton adhesiveness to the lipid bilayer. While RBCs from hereditary elliptocytosis patients retain their adhesiveness, the adhesiveness of RBCs from hereditary spherocytosis patients is significantly reduced. Therefore, the role of membrane constituents in ensuring membrane adhesiveness can be weighed by modeling the behavior of these two cell types as they pass through IES and, specifically, by analyzing the process of vesicle formation. Li and colleagues [102,104] used a two-component protein-scale cell model to simulate the passage of IES by a red cell to accomplish this task. They concluded [102] that the spectrin-deficient RBCs are more amenable to vesiculation than the RBC with deficient band 3 (anion exchanger protein AE-1) when they traverse IES. These results are also significant from a clinical point of view, as they explain why splenectomy prolongs survival in the case of spectrin/ankyrin-deficient RBCs but not in the case of band 3-deficient cells [105].

Garcia-Herriros et al. [35] demonstrated that the total number of vesicles increased with an increasing number of times the cell passed through a constriction and reached a plateau at around 60 cycles, which coincides with the number of cycles after which the mean corpuscular volume (MCV) of the cells stopped decreasing.

It is important to emphasize that the vesiculation process modulates changes in some RBC properties [54,106,107], and vesicle concentration may indicate the level of RBC damage following exposure to MS [108]. From this point of view, Wei et al. [30] have concluded that “RBCs modulate their morphologies to maintain the biconcave shape by losing more cytoplasm and leaving a larger redundant surface area”. This vesicle formation is linked to the depletion of membrane lipids and proteins, including stomatin and flotillin, which play a critical role in modulating membrane viscoelasticity and diminishing membrane deformability [18,109,110].

### 3.3. RBC Membrane Composition/Structure

As was shown previously for various RBC states (such as aging, oxidative stress, storage, etc.), vesiculation is accompanied by the loss of some proteins and lipids from the cell membrane [110]. Thus, the cell membrane loses several proteins during aging [111,112]. Some of them (band-3, flotillin-1, stomatin, protein 4.1, ezrin, flotillin-2, and argonaute-2) are an integral part of the vesicles formed by the cell [110,113,114,115] so that vesiculation is associated with a decrease in the level of proteins (including flotillin-2, flotillin-1, and stomatin) in the membrane lipid rafts [116,117,118,119]. In addition, microvesicles may contain high contents of glucose transporter 1 (GLUT1), band-3 (AE1), aquaporin-1, and CD-47, which have been recently identified in the RBC membrane as integral parts of the complexes established by stomatin with other proteins [120]. Accordingly, we may expect a decrease in the number of protein fractions in RBCs exposed to MS due to their vesiculation (see Section 3.2).

Garcia-Herreros et al. [35] subjected RBCs to oscillatory MS as they passed through successive constrictions. The authors reported a change in several cellular features, particularly RBC vesiculation and loss of the same proteins from the membrane. In addition, they focused on examining the distance between two neighboring molecules of band-4.1 or ankyrin-1. For this purpose, the inter-protein distances were determined for cells that passed through 200 constrictions using stimulated emission depletion (STED) microscopy and image correlation analysis. The distance between both membrane proteins (band-4.1 and ankyrin-1, main bilayer–cytoskeleton cross-linkers) was reduced after cell exposure to MS, which is known to impact the complex’s stability. Moreover, the effective bending modulus of the bilayer–cytoskeleton complex increased with cyclic mechanical loading. Based on these results, the authors concluded that vesiculation depletes several proteins under conditions of MS, while most other proteins are slightly enriched due to cell compaction. The observed change in membrane structure (the bilayer–cytoskeleton linker network becoming denser) is consistent with the increase in the effective bending modulus reported in the same publication [35]. Thus, the authors’ [35] main claim is that as the vesicles are released, the membrane’s distance between the bonds (skeleton and lipid layer) gradually decreases, limiting the bilayer oscillation.

It should be noted that the authors [35] did not record a decrease in the stomatin and flotillin content in the membrane following vesicle formation caused by RBC aging [110,118,121]. This may indicate that the mechanisms of stress-induced vascularization differ from those of cell aging (in vivo and in vitro).

### 3.4. RBC Morphology

As mentioned above, RBC vesiculation and changes in cell morphology are connected by a cause-and-effect relationship [108,122]. Wei et al. [30] established the occurrence of significant morphological changes in the cell after the application of stress oscillations. The authors [30] used a microfluidic system with a contraction region in which RBCs are repeatedly subjected to stretch and relaxation. Experimental results demonstrated that the exposure of RBCs to mechanical stress led to the loss of RBC surface area and, as a result, to cell shape transformations. Based on their results, Wei and colleagues [30] developed a mathematical model for estimating the evolution of the cell surface area and the membrane shear modulus under shear stress.

Qiang et al. [58], who also used microfluidic devices with alternating contraction and expansion during flow, pointed out that oscillatory MS induces damage to RBCs after a critical number of cycles (contraction and expansion) and accumulates over hundreds of fatigue cycles. Following cyclic exposure, the authors observed changes in the RBC shape from a discoid to an elliptocytic and a stomatocytic shape. In addition, in several cases, the development of spike-like protrusions on the surface of the cell membrane was observed.

Pan et al. [28] propose both a mechanism and a timeline for the membrane structural changes by dividing the process of erythrocyte deformation under oscillatory stress conditions into three stages following changes in morphology. The first stage is characterized by preserving the RBC’s ability to elastically recover after passing an obstacle. At this stage, most RBCs retain their original biconcave shape. Dramatic changes in morphology characterize the second deformation stage; as the number of contractions and expansions increases, echinocytes and spherocytes form. At the third and final stage of fatigue, a lysis of compressed, deformable spherocytes occurs. Furthermore, the authors consider RBC periodical compression to be the main factor that leads to changes in the morphology of the cells [28]. Finally, they link the changes in the morphology to the disruption of the band 3-ankyrin bonds between the cell membrane and the skeleton, which is caused by the depletion of the ATP stores due to periodic compression of the cells passing through obstacles. Eventually, changes in cell morphology cause the deformability of the squeezed RBCs to decrease, and Young’s modulus increases [28].

### 3.5. RBC Deformability

Deformability is one of the vital features of RBCs, and it is susceptible to exposure to mechanical stress. Healthy cells deform almost elastically in response to shear stress in circulation, facilitating their efficient passage through capillaries and sinusoids of the spleen, which are narrower than the RBC diameter. RBCs with reduced deformability (increased rigidity) impair perfusion and O_2_ delivery to peripheral tissues [56,123,124,125] and can directly block capillaries [56,126,127]. It is important to note that the distribution of erythrocyte deformability in the cell population is not uniform; along with highly deformable cells, there are also undeformable and low-deformable RBCs. Undeformable and low-deformable (rigid) RBCs prevent the passage of the cells through the spleen vasculature and increase splenic RBC sequestration and destruction [14,102,128]. For this reason, an alteration in RBC deformability induced by MS (for example, during perioperative blood salvage [129] or circulation in artificial organs [62]) has clinical significance.

Cell deformability is one of the main properties that characterize cell sensitivity to MS exposure [11,32,71,130,131,132]. Baskurt et al. [44,130] studied this phenomenon by subjecting RBCs to constant shear stress (120 Pa) for 15–120 s. Under these conditions, RBC deformability was substantially decreased without significant hemolysis. Dao et al. [133] subjected cells to a shear stress of 100 Pa for 120 s and found decreased cell filterability.

The change in RBC deformability is typically related to altered cell morphology, membrane viscoelasticity, and an increase in the viscosity of the cytosol [134,135,136]. As described above, cell morphology and the level of skeleton attachment to the lipid bilayer are altered significantly during cells’ exposure to MS, which can explain the change in deformability.

As described above, RBCs can be subjected to MS exposure under static conditions [13,21,31,59] or under cyclic (repeated contraction and expansion) [10,35,52] impact while circling in the human body or artificial organs. In modeling experiments (in vitro), RBCs gradually lose their deformability when they flow through a repeated contraction in a microfluidic device [57], are subjected to periodic electrical deformation [58], or are cyclically deformed using a Couette shearing system [13,21,27,52]. Moreover, it has been shown that the loss of RBCs’ deformability during cyclic deformation is much faster than that occurring under static conditions at the same maximum load over and accumulated duration of exposure [58]. This MS-induced loss of deformability is more marked at higher amplitudes of repeated constrictions and expansion [58].

Horobin et al. [21,137] studied the alteration of RBC deformability under sub-hemolytic conditions under constant MS. For that purpose, the authors subjected RBC suspensions to short-term (1–16 s) shear stress (from 5 up to 100 Pa) in an ektacytometer (LORRCA MaxSis, Mechatronics, The Netherlands). The authors demonstrated that the cell response to MS depends on the magnitude of the stress and the exposure duration. Moreover, the authors of several other studies suggested that specifically, repeated compression exerts the most significant influence on cell deformability [10,58].

McNamee et al. [13] used a counter-rotating shear generator mounted on a microscope [34,138] to visualize the morphology of deformable RBCs subjected to a flow under supra-physiological shear stresses (10–60 Pa). The authors demonstrated that while the cell elongation index (aspect ratio) remained unchanged under all conditions, the frequency of asymmetric ellipses and erythrocyte/extracellular vesicle fragments formation increased significantly after RBC exposure to shear stress of 60 Pa for 100 s (ms). Based on their results, they concluded that asymmetric cell morphology may indicate sublethal blood injury [13].

In a previous study, we examined the altered deformability of RBCs exposed to MS generated during mechanical oscillation in a bead mill [22], during the preparation of RBC samples from donated blood [17], and during peri-operative blood salvage [129]. Specifically, we have demonstrated that an increase and a decrease in the percentage of undeformable cells (%UDFC) can occur after exposure of RBCs to the above conditions [17,18,129]. We also demonstrated that the change in %UDFC depended on the initial size of undeformable cell fraction: it increased when the baseline %UDFC was low and decreased when it was high. Based on the data obtained, we [17,18,129] concluded that it is likely that the exposure of RBCs to high MS during the preparation of packed RBC (PRBC) unit produces two opposite effects: first, the hemolysis of rigid (undeformable) cells, thereby reducing their percent (i.e., the % UDFC); second, mechanical damage to the RBC membrane with subsequent reduction in the cell’s ability to deform, thereby increasing the %UDFC.

A possible explanation for this phenomenon can be found in the studies of Sakota et al. [45] and Yokoyama et al. [139], demonstrating that the flow of RBCs induced by rotary pumps has (leads to) an increased mean corpuscular volume (MCV) and a decreased mean corpuscular hemoglobin concentration (MCHC). Yokoyama et al. [139] speculated that “the likely mechanism is that older RBCs with smaller size and higher hemoglobin concentration were destroyed faster by shear stress, while younger RBCs with larger size and lower hemoglobin concentration can hold on”. Thus, they assumed that the selective destruction of the aged RBCs occurred by removing more fragile cells with a lower hemolysis threshold. Due to the aged RBCs’ higher fragility [140] (i.e., lower resistance to MS), their selective destruction results in an increase in the average MCV and a decrease in the average MCHC, as reported by both Sakota and Yokoyama [45,139].

Summarizing the results of research conducted by Sakota et al. [45], Yokoyama et al. [114], and our group [17,18,129], we conclude that MS has a double-faceted effect on RBCs; the most aged RBCs (not capable of maintaining their integrity) undergo hemolysis. At the same time, the properties of the remaining intact cells are subjected to significant changes.

### 3.6. RBC Fragility

Red cells are damaged or destroyed when the shear stress applied to them exceeds certain limits. RBCs’ mechanical fragility refers to the susceptibility of erythrocytes to hemolysis under these conditions [141]. The mechanical fragility of RBCs is a critical variable for evaluating hemolysis caused by multiple necessary clinical devices, such as pumps, valves, gas exchange devices, cannulae, etc.

Structural changes within the cells accumulate during exposure to MS [28,30,57,58,69]. This “cellular trauma” decreases their resistance to mechanical and osmotic stress, making them more fragile. Kamenenva and her colleagues [32] compared the level of fragility of three types of cells subjected to MS: young, old (obtained using density separation by Percoll gradient), and native RBCs. Mechanical stress was created in a cup piston apparatus [142]. The authors found that the exposure of blood to MS increases the mechanical fragility of RBCs, including their partial hemolysis. Moreover, the authors also documented a similarity between cells exposed to MS and in vivo aged RBCs. Inoue et al. [42] and Yazer et al. [141] reached similar conclusions when they studied the effect of MS generated by blood pump [42] and intraoperative auto-transfusion [141] on RBCs’ mechanical fragility. Inoue et al. [42] used porcine blood samples circulated by a blood pump (BP-80) at the beginning of circulation and 3 h afterward. During the experiments, a significant increase in cell fragility was demonstrated after only three hours in circulation [42].

### 3.7. The Ability of RBCs to Form Aggregates and Their Adhesiveness to Endothelial Cells

Under physiological conditions, two types of intercellular interactions are characteristic of RBCs in the bloodstream: aggregation (adhesion between two cells) and adhesion to endothelial cells. The main factors influencing the intensity of these interactions are RBC surface charge and plasma protein composition [143,144,145,146,147,148,149]. Since MS can cause a decrease in the surface charge, one should expect a change in cells’ aggregability and adhesiveness following the application of mechanical stress.

The ability to form multi-cellular aggregates depends on the presence of plasma proteins [150,151]. Opposite forces determine the extent of aggregation: the repulsive force between the negatively charged cells [145], the cell–cell cohesion induced by plasma proteins, and the flow-induced disaggregation resulting from shear stress [152,153,154]. At normal physiological conditions, the flow-induced shear stress is sufficient to cause disaggregation before cells pass through capillaries. However, pathological conditions such as inflammatory states and oxidative stress significantly influence cell aggregation, impacting the blood flow [146,148,155,156,157]. The increased ability of cells to form aggregates is associated with cardiovascular diseases [158] and correlates well with inflammatory indices of patients with cardiovascular conditions and sepsis [146].

McNamee et al. [159] demonstrated that the aggregability of RBCs increased after application of shear stress of 125 Pa for 1.5 sec. As RBC surface charge has a strong impact on erythrocytes’ aggregation [143,144], this phenomenon can be attributed to the loss of surface sialic acid [12,32,159] caused by MS exposure. Freitas Leal et al. [33] examined two parameters (aggregation index and threshold shear stress) that characterize RBC aggregability and aggregate strength in patients on extracorporeal devices. The authors demonstrated that both parameters increased to a stable level during circulation-assisted surgery [33].

RBC adherence to endothelial cells (EC) lining the walls of the blood vessel (hereafter “adherence”) can promote occlusion of microvessels [160]. Typically, RBC adherence to EC is clinically insignificant, but under oxidative stress it is enhanced [155], and contributes to microcirculatory disorders observed in various pathologies, particularly those associated with oxidative stress, such as sickle cell disease, malaria, and thalassemia [137,161]. Thus, Kaul et al. [162] demonstrated that the perfusion of red cells with elevated adherence caused a substantial elevation of vascular resistance in rat mesocecum.

Vijayaraghavan et al. [47] studied the sensitivity of RBC adhesiveness to MS exposure. The authors subjected RBCs to shear stress of 0.53 up to 4.05 Pa for 15–60 s in a parallel flat chamber. Frangos et al. described this chamber’s construction in detail [163]. Vijayaraghavan et al. [47] demonstrated that elevation of applied shear stress increased RBC adhesion to cultured endothelial cells. Thus, adhesive cells increased fourfold after one minute of RBC exposure to shear stress of 4.5 Pa.

### 3.8. Phosphatidylserine Externalization

The RBC membrane has an asymmetric distribution of phospholipids across the bilayer. The outer leaflet of the RBC plasma membrane is significantly enriched with the choline phospholipids (sphingomyelin and phosphatidylcholine). In contrast, the majority of the amino-phospholipids (phosphatidylserine (PS) and phosphatidylethanolamine (PE)) are confined to the membrane’s inner leaflets [164]. The asymmetric distribution of the phospholipids across the RBC membrane is based on the cooperative activities of three transporters: ATP-dependent amino-phospholipid translocase (flippase), which rapidly transports PS and PE from the cell’s outer-to-inner leaflet; ATP-dependent nonspecific lipid floppase, which slowly transports choline-phospholipids from the cell’s inner-to-outer leaflet; and Ca^2+^-dependent nonspecific scramblase, whose activity leads to the lateral diffusion of phospholipids between both leaflets [165]. Moreover, the externalization of PS can be activated due to band-3 clustering [166].

PS externalization to the RBC surface is an essential regulatory factor in blood cell physiology and circulation. For example, surface PS causes the catalytic activity of various proteins involved in blood coagulation, which can induce platelet activation [167] and hypercoagulation [168] in pathological states such as thalassemia [169] and sickle anemia [170]. PS externalization has recently been proposed as a major cause of erythrocyte phagocytosis by macrophages, which have a specific scavenger receptor recognizing surface PS [171]. The increased RBC/EC interaction seems especially pronounced in pathologies characterized by a subpopulation of PS-exposing RBC, such as sickle cell anemia [172], diabetes [173], cerebral malaria [174], and thalassemia [175]. The exposure of PS on the surface of RBCs may be one of the main mechanisms of their adherence to ECs. Some works demonstrated a linear correlation between the number of PS-exposing RBCs and the number of RBCs adhering to the ECs [176,177]. Another work has proposed that activated ECs express adhesion receptors that directly interact with PS at the outer surface of RBCs without plasma proteins [178].

Garcia-Herreros et al. [35] conducted an extensive study of the effects of MS on various cell features, particularly the level of PS externalization. To this end, the authors simulated the conditions that exist when red cells pass through the IES. The study tested the relationship between the number of RBC passages through the constriction and the percentage of PS-exposing cells in the population. The authors detected increased PS exposure only after a high deformation cycle number. Interestingly, if a significant level of hemolysis was observed already at the 64th cycle (repeated flow through constrictions), PS-externalization became noticeable only after the 480th cycle. These results contrast significantly with the data presented by Freitas Leal et al. [33], which, while obtained under entirely different conditions of MS exposure, demonstrated the high sensitivity of RBC PS asymmetry to mechanical exposure.

The latter work [33] studied the changes detected in the concentration of PS-exposing RBCs during their circulation in a heart–lung machine. To this end, the authors examined RBCs stored in the blood bank for 1, 3, and 5 weeks. The fraction of PS-exposing RBC in the general population of cells was elevated during the first 2.5–3.0 h of circulation in the stand-alone extracorporeal circuit [33]. A light decrease in the indicator was observed in the following hour of circulation. During surgery, the patient’s RBCs show a similar pattern: an initial increase followed by a slight decrease. Thus, the data obtained from the analysis of stored blood can be extrapolated to explain the increase in RBC adhesion observed during surgical intervention.

### 3.9. RBC Lifespan

As described above, exposure of RBCs to MS causes a decrease in cell deformability and surface charge, accompanied by externalization of PS. All these factors are markers of RBC aging; therefore, it can be assumed that the lifespan of cells repeatedly subjected to MS should be negatively affected.

Indeed, several studies conducted on patients and animal models have confirmed the suggestion that cell lifespan is reduced under mechanical stress. The authors of these studies compared the lifespan of RBCs collected from prosthetic heart valve (PHV) patients with that of healthy controls. In these patients, the blood is subjected to supra-physiological shear stress [71,72,73,74], which causes a decrease in erythrocyte deformability. This assumption was confirmed by modeling fluid flow dynamics through artificial PHV. Several authors have demonstrated by numerical [179,180] and laboratory [181,182] methods that flow turbulence occurs when blood passes through PHV and can cause RBC damage [65,183]. Consistent with these results, patients with artificial heart valves are characterized by a shorter RBC lifespan [64,65]. Thus, Mitlyng and colleagues showed that the RBC circulation span in patients with PHV is only 99 days instead of the expected lifespan of about 120 days [184].

Another type of patient in whom the effects of mechanical stress on RBC lifespan have been extensively studied is the patient undergoing various types of dialysis [66,67]. Among others, Luo et al. [67] studied the effect of hemodialysis on RBC lifespan in patients with end-stage kidney disease. The authors concluded that dialysis using a polysulfone membrane does not appear to disrupt RBCs or reduce their lifespan. Vos et al. [66] measured the lifespan of RBCs in long-term dialysis patients and detected a decrease in cell survival. However, the authors do not connect the observed effect with MS applied during dialysis [185].

## 4. Possible External Factors and Mechanisms of the Impact of the Sensitivity of RBCs to Mechanical Stress

### 4.1. Role of Extracellular Fluid on the Sensitivity of RBCs to Mechanical Stress

It is well known that macromolecules or proteins in the surrounding fluid primarily determine some properties of the RBCs [148,149,153,186]. In our previous research, we examined the role of medium-related factors in RBC behavior. For example, adding dextran to the protein-free buffer leads to RBC aggregation [153], and supplementation of PBS by albumin leads to depression of nanoparticle hemolytic activity [187]. Similarly, it has been shown that RBC adhesion to EC in plasma is more than an order of magnitude higher than that in PBS buffer [17].

The RBCs are permanently suspended in plasma, whether exposed to MS in blood vessels or artificial organs. Numerous studies have demonstrated that proteins in the extracellular fluid significantly reduce MS-induced hemolysis [46,70,71,188]. The first study analyzing the role of plasma components in RBC mechanical trauma was published by Kamada et al. [188]. The authors demonstrated that patients on artificial circulation experienced changes in RBC morphology, while the albumin administration attenuated this effect. It was later shown that plasma proteins contributing to the protective effect are albumin, warmed plasma supernatant, and a heat-stable extract of fresh-frozen plasma (FFP) [70,189]. In contrast, γ-globulin, haptoglobin, ceruloplasmin, and α-2-macroglobulin provide little protection [189]. Sumpelman and colleagues [190] speculated that this protective effect should be attributed to negatively charged proteins that attach reversibly to the RBC membrane. Later, Kameneva et al. [46] demonstrated that blood supplementation with polyethylene glycol (PEG) reduces MS-related hemolysis. The authors proposed that the protective mechanism of PEG may result from coating the cell surface with the artificial material and from absorption of PEG into the RBC membrane complex, thereby increasing the resistance of RBCs to shear stress [46]. Later, several authors suggested that plasma proteins coat the surface of RBCs to help seal the membrane ruptures and correct the damage inflicted on the cell membrane by MS [46,189].

### 4.2. Possible Mechanisms of the Impact of Mechanical Stress on the Composition/Structure of the RBC Membrane

Shear stress causes fluctuations in the membrane of RBCs as they undergo deformation. These fluctuations are more pronounced under oscillatory shear stress than under constant stress. Such fluctuations affect the interactions among the rearranged band 3 molecules, spectrin conformational alterations, and lipids’ reorganization within the lipid bilayer.

The alterations in band 3 molecules elevate the proportion of free band 3 molecules from the typical 30% observed in intact erythrocytes to approximately 60%. This increase is attributed to the disintegration of low-affinity band 3–adducin complexes, as evidenced by a dissociation constant of about 100 nM [191,192]. Band 3 complexes with ankyrin and adducin, alongside free band 3 molecules, are illustrated schematically in Figure 1. In contrast, high-affinity band 3-ankyrin complexes remain stable despite these external fluctuations. The resulting conformational changes enhance electrostatic interactions between the highly anionic N-terminal domains of band 3 molecules, thereby increasing their tendency for self-association [80]. The freely diffusing band 3 molecules can form clusters and dynamically attach to or detach from spectrin, impacting the persistence length of spectrin molecules and inducing conformational changes that affect their flexibility.

The flexibility of spectrin (see Figure 2), which ranges from entirely flexible to semi-flexible [84,104], is affected by two primary factors: (1) the quantity of band 3 molecules associated with individual spectrin filaments and (2) the phosphorylation status of the actin junctions. The cumulative effects of modified conformational changes in spectrin, which arise from (i) external perturbations and (ii) the reorganization of band 3 molecules, exert a reciprocal influence on the viscoelastic properties of the cytoskeleton. Viscoelasticity, in this context, describes the relationship between energy storage and the structural alterations of the membrane [193].

Alterations in the viscoelastic properties of the cytoskeleton, which occur alongside the reorganization of band 3, significantly affect the coupling between the cytoskeleton and the lipid bilayer. The restructuring of band 3 initiates changes in the bilayer structure. As rigid inclusions, band 3 molecules contribute to localized bending and compression of the adjacent lipid bilayer. These structural modifications within the bilayer subsequently influence the positive hydrophobic mismatch effects between band 3 proteins and lipids, potentially leading to protein tilting [82,193,194,195]. The tilting of the protein is driven by both long-range interactions mediated by lipid curvature and short-range lipid compression [196]. Furthermore, the reorganization of lipids affects the bending of the bilayer, which in turn has a reciprocal effect on the viscoelasticity of the cytoskeleton.

Cholesterol molecules play a crucial role in stabilizing the bending states of the lipid bilayer [197,198,199]. According to Leonard et al. [198], domains enriched with cholesterol tend to accumulate in regions of high curvature resulting from cellular deformation. The lateral diffusion coefficient of cholesterol has been shown to diminish as the local concentration of cholesterol increases. Additionally, the presence of the trans-membrane protein band 3 further restricts the mobility of cholesterol [200]. As a result, the spatial arrangement of band 3 molecules significantly impacts the lipid bilayer’s bending state. In contrast, the clustering of band 3 is associated with the externalization of PS, with modification of the viscoelastic properties of the membrane, and with increased fragility of red blood cells [166].

Conformational alterations in band 3 molecules and lipid rearrangement induced by shear stress impact the organization of hemoglobin [92]. The deformability of cells depends on the membrane viscoelasticity. In this context, non-linear viscoelasticity is influenced by several factors: (i) the viscoelastic properties of the cortex, which depend on the flexibility of spectrin filaments; (ii) the bending elasticity of the bilayer; and (iii) the interaction between the bilayer and the cortex—viscoelasticity results from both energy storage and energy dissipation that occur during membrane fluctuations. The storage modulus quantifies the energy stored, while the loss modulus, derived from microrheological experiments, quantifies the energy dissipated. A representative set of experimental data are illustrated schematically in Figure 3, inspired by the experimental data from Amin et al. [201].

Two distinct viscoelastic regimes were identified. In the regime characterized by lower angular velocities, the condition is met in which energy storage exceeds energy dissipation, as shown in Figure 3. Conversely, this relationship is reversed in the regime associated with higher angular velocities. The lower angular velocity regime leads to structural alterations in the membrane at a supramolecular level. In this context, energy storage is mainly attributed to the reorganization of mesoscopic domains within the cortex and the interaction between the cortex and the bilayer. In contrast, the significant energy dissipation observed at elevated angular velocities results from conformational changes in spectrin filaments, the partial disruption of band 3 clusters, and the lateral diffusion of lipids.

It is essential to consider various forms of hemoglobin–membrane interactions, which include (a) the electrostatic attachment of deoxyhemoglobin to the cytoplasmic region of band 3 [202,203], (b) the covalent crosslinking with membrane constituents through disulfide bonds, and (c) the adsorption to membrane lipids mediated by hydrophobic interactions [204,205]. Increasing the number of hemoglobin molecules bound to the membrane affects its viscoelastic properties, leading to enhanced stiffness, reduced deformability, and increased fragility.

## 5. Future Perspectives

As demonstrated above, various research groups have gathered experimental data on cells’ sensitivity to mechanical stress. However, this information is fragmented and was not collected under standardized conditions, making comparing findings from different authors challenging. Additionally, researchers have overlooked some issues that cannot be resolved due to insufficient accumulated data.

For example, although nearly 500 publications (according to the data presented in PubMed) have studied the sensitivity of RBCs to the effects of MS, the role of cellular and plasma factors in this process has either not been analyzed or has been analyzed insufficiently. Most researchers have focused their attention on the role of the exposure conditions (their strength, duration, and mode of action) in determining the outcome of exposure but have not considered the effect of the initial RBC state on its sensitivity to MS. The only exceptions are two studies conducted by Sakota et al. [45] and Yokoyama et al. [139], and our group [17,129], in which an attempt was made to link the initial properties of the red cells with the magnitude of changes occurring as a result of exposure. In this regard, there is no clear understanding as to which cellular factors determine the sensitivity of red blood cells to MS. This situation is, by necessity, related to the fact that we still do not have a comprehensive molecular model of this phenomenon.

It should also be noted that studies devoted to the effect of extracellular fluid composition on RBCs’ sensitivity to MS mostly focused on cell hemolysis, while the impact of environmental factors on the resistance of other cell features to mechanical stress have not been sufficiently addressed. In this regard, we do not yet have an answer as to why the presence of proteins in the extracellular fluid reduces the sensitivity of cells to mechanical stress.

This section will briefly debate future research directions in the area under discussion.

Most previous studies dealing with the effects of MS on RBCs focused on MS-related hemolysis [10,24,48,51], the extreme outcome of MS exposure. However, obtaining a clear understanding of pre-hemolytic mechanical damage is also essential. However, some research groups (including ours) have previously investigated changes in various RBC properties resulting from MS exposure, with each group concentrating on a narrow range of features. Such a fragmented approach does not allow us to obtain a general picture of what is happening to the RBCs under mechanical stress and define parameters most sensitive to exposure. For this reason, conducting a comprehensive study that will involve scientists with various areas of expertise (apparently belonging to different research groups) is necessary. Such a consortium will allow us to implement a combination of biochemical, biophysical, hemodynamic, and numerical research methods and to consider the various questions from different angles in an attempt to establish the following:(a)The relationship between RBC’s initial biochemical/biophysical features and cell sensitivity to mechanical exposure (in other words, which features of native RBCs best determine their resistance to MS);(b)The role of extracellular fluid on RBC sensitivity to mechanical exposure (such as investigating why the presence of proteins in the environment reduces the sensitivity of RBCs to MS);(c)The changes occurring in RBC features following packed red cell unit preparation or during peri-operative blood salvage (surprisingly, this issue has not received enough attention from researchers in the field despite its enormous relevance, with approximately 30,000 units of packed RBCs being supplied daily in the United States alone);(d)The relationship between MS-induced changes in the RBC membrane composition/structure and alteration in cell functionality (aiming to delineate the correlation between the change in the content of membrane proteins and the observed MS-related alteration in the RBCs properties and to create a theoretical model explaining this correlation).

Of course, multiple additional questions will arise as the research progresses, requiring the development of new tools and theoretical models.

## 6. Conclusions

To summarize, the impact of mechanical stress on cell properties is widely discussed in the scientific literature—the authors of the cited publications employed various research methods to characterize the phenomenon and clarify its mechanisms. Overall, we can conclude that RBCs experience significant changes when subjected to mechanical stress during blood circulation in the body or through artificial organs [12,13,15,20,34,45,51,206] and that this impact leads to accelerated cell aging [35]. Moreover, the exposure of RBCs to mechanical stress may ultimately lead to hemolysis [10,24,26,46,48,51,53,65,141,142,207,208] by various mechanisms suggested by experimental and numerical analysis [26,48,142]. However, despite the extensive information gathered by multiple research teams, several questions remain unanswered, and a comprehensive model of RBC sensitivity to MS has yet to be established. The authors of this review hope that these issues will be addressed in future research.

## Figures and Tables

**Figure 1 micromachines-16-00259-f001:**
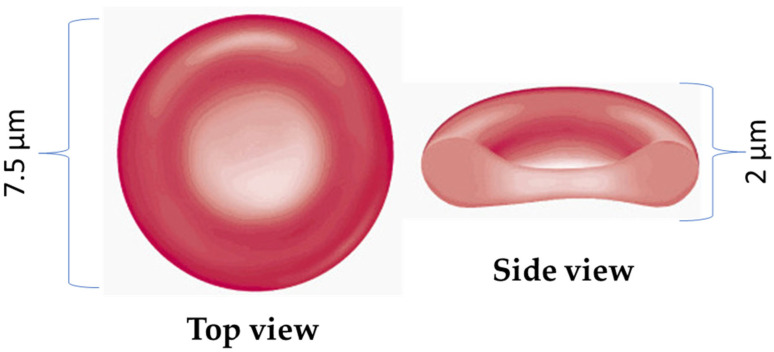
**Schematic representation of red blood cells cell.** Normal RBCs are flexible and disk-shaped, thicker at the edges than in the middle. The RBC membrane forms its surface, and inside the cell is the cytosol, an aqueous solution of hemoglobin containing various salts and proteins.

**Figure 2 micromachines-16-00259-f002:**
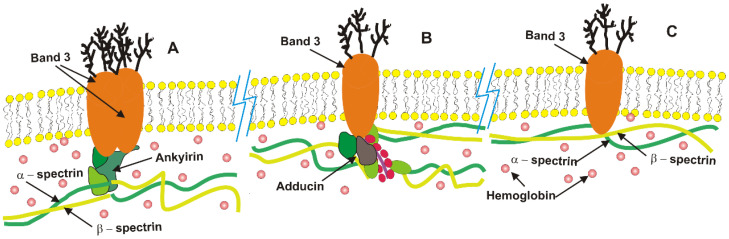
**A schematic representation of the RBC membrane structure, highlighting its key functional components**: (**A**) high affinity band 3-ankyrin complexes located near the center of spectrin tetramers, (**B**) lower affinity band 3-adducin complexes, and (**C**) free band 3, which move laterally along the membrane by establishing short-lived unstable complexes with spectrin. The majority of hemoglobin molecules diffuse within the cytosol, while some of them are attached to the lipid bilayer and protein complexes.

**Figure 3 micromachines-16-00259-f003:**
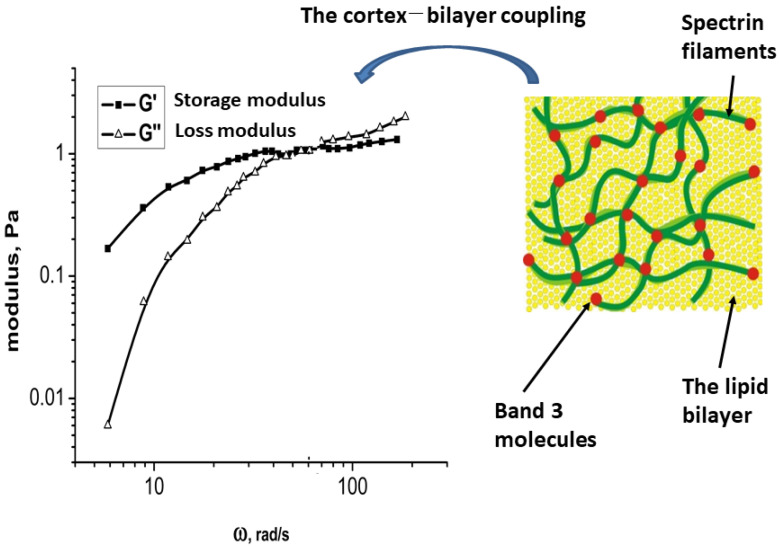
The viscoelastic properties of the erythrocyte membrane are assessed by analyzing the energy storage and dissipation in relation to angular velocity, which results from the structural alterations of the membrane components during fluctuations. This viscoelastic behavior can be examined across two distinct regimes of angular velocities (or time scales).

**Table 1 micromachines-16-00259-t001:** Devices prevalent in clinical settings or specifically designed to simulate mechanical stress.

№	Device	Conditions of Exposure	Reference
1	Microfluidics	Repeated constrictions	[32,34,37,39,61,62]
2	Microfluidics	Laminar flow	[40,63]
3	Needles and catheters	Laminar flow	[64,65]
4	Blood pump	Circulation flow	[46,49,66]
5	Prosthetic heart valves	Turbulent flow	[67,68,69]
6	Parallel flat chamber	Laminar flow	[47]
7	Couette shearing system	Laminar flow	[17,25,31,56]
8	Hemodialysis device	-	[70,71]
9	Electrodeformation	Cyclic deformation	[72,73]
10	Bead mill	Mechanical oscillation	[22,36,50,74,75,76]

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
