# Peer review of "Mechanical Stimulation of Red Blood Cells Aging: Focusing on the Microfluidics Application"

_micromachines, 2025, doi:10.3390/mi16030259_

Round 1
Reviewer 1 Report
Comments and Suggestions for Authors
The manuscript addresses an important topic by examining the response of red blood cells (RBCs) to varying levels of shear stress, with their physiological and pathological behavior. The authors systematically described the existing knowledge, which are clearly defined, and the focus on microfluidics is valuable, as the microfluidic approach is a state-of-the-art approach currently for blood flow simulations. Additionally, the inclusion of diverse RBC properties, such as the cytosol, cytoskeleton, and membrane, indicates a thorough and comprehensive review. I believe the review manuscript will fill an important gap in the literature and elevate the level of understanding in a critical area.
Author Response
Thanks to the reviewer for the positive assessment of our review.
Reviewer 2 Report
Comments and Suggestions for Authors
The review examines a very interesting problem about the response of erythrocytes to mechanical impact. The review is compiled by world-renowned experts in this field. I read this review with great interest and am grateful to the authors for drawing my attention to many important details
This is an excellent review. I will recommend the review for publication.
However, the review requires main revision.
I am sure that this review will be of interest to a wide range of readers.
For this reason, I ask the author to add one (or even two) drawings that will schematically illustrate the erythrocyte.
The authors should show many important details such as lipid bilayer, cytoskeleton, spectrin, hemoglobin, etc.
The authors should add a brief educational subsection describing the functionality of the erythrocyte from a mechanical point of view. This should be an introduction to physiology for mechanical engineers and an introduction to mechanics for physiologists.
My little confusion.
Line 435:
Another type of patient in whom the effects of MS have been extensively(?) studied is the…
What does (?) mean here?
Author Response
Thanks to the reviewer for the positive assessment of our review.
I am sure that this review will be of interest to a wide range of readers.
For this reason, I ask the author to add one (or even two) drawings that will schematically illustrate the erythrocyte.
The authors should show many important details such as lipid bilayer, cytoskeleton, spectrin, hemoglobin, etc.
The authors should add a brief educational subsection describing the functionality of the erythrocyte from a mechanical point of view. This should be an introduction to physiology for mechanical engineers and an introduction to mechanics for physiologists.
Thanks to the reviewer for his suggestion. Following the comment, we have added a new section (Section 2) to the manuscript's text (lines 129 - 226) in which we have tried to describe the topic indicated by the reviewer. In addition, at the reviewer's suggestion, we have added a relevant Figure to the second section.
My little confusion.
Line 435:
Another type of patient in whom the effects of MS have been extensively(?) studied is the…
What does (?) mean here?
We appreciated the reviewer's attention to the typo, so we made the necessary changes (lines 564-565) to the new version of the manuscript.
Round 2
Reviewer 2 Report
Comments and Suggestions for Authors
The authors complied with my request and added an explanatory picture and a short educational subsection.
I hope this only improved the review.
Author Response
We are grateful to the reviewer for his positive assessment of our manuscript.